# Differential Factors for Predicting Outcomes in Left Main versus Non-Left Main Coronary Bifurcation Stenting

**DOI:** 10.3390/jcm10143024

**Published:** 2021-07-07

**Authors:** Jung-Joon Cha, Soon Jun Hong, Hyung Joon Joo, Jae Hyoung Park, Cheol Woong Yu, Tae Hoon Ahn, Hyo-Soo Kim, Woo Jung Chun, Seung-Ho Hur, Seung Hwan Han, Seung-Woon Rha, In-Ho Chae, Jin-Ok Jeong, Jung Ho Heo, Junghan Yoon, Ki Hong Choi, Young Bin Song, Hyeon-Cheol Gwon, Jong-Seon Park, Myeong-Ki Hong, Joon-Hyung Doh, Kwang Soo Cha, Doo-Il Kim, Sang Yeub Lee, Kiyuk Chang, Byung-Hee Hwang, So-Yeon Choi, Myung Ho Jeong, Chang-Wook Nam, Bon-Kwon Koo, Do-Sun Lim

**Affiliations:** 1Department of Cardiology, Cardiovascular Center, Korea University Anam Hospital, Korea University College of Medicine, Seoul 02841, Korea; joonletter@hanmail.net (J.-J.C.); drjoohj@gmail.com (H.J.J.); jhpark3992@naver.com (J.H.P.); ycw717@naver.com (C.W.Y.); ath3869@naver.com (T.H.A.); dslmd@kumc.or.kr (D.-S.L.); 2Department of Internal Medicine and Cardiovascular Center, Seoul National University Hospital, Seoul 03080, Korea; hyosoo@snu.ac.kr (H.-S.K.); bkkoo@snu.ac.kr (B.-K.K.); 3Division of Cardiology, Department of Internal Medicine, Samsung Changwon Hospital, Sungkyunkwan University School of Medicine, Seoul 51353, Korea; saintjmn@naver.com; 4Division of Cardiology, Department of Internal Medicine, Keimyung University Dongsan Medical Center, Daegu 42601, Korea; babycat81@hanmail.net (S.-H.H.); ncwcv@dsmc.or.kr (C.-W.N.); 5Division of Cardiology, Department of Internal Medicine, Gachon University Gil Hospital, Incheon 21565, Korea; kheartist@gmail.com; 6Division of Cardiology, Department of Internal Medicine, Korea University Guro Hospital, Seoul 08308, Korea; swrha617@yahoo.co.kr; 7Division of Cardiology, Department of Internal Medicine, Seoul National University Bundang Hospital, Seongnam 13620, Korea; ihchae@snu.ac.kr; 8Division of Cardiology, Department of Medicine, Chungnam National University Hospital, Daejeon 35015, Korea; jojeong@cnu.ac.kr; 9Division of Cardiology, Department of Internal Medicine, Kosin University Gospel Hospital, Kosin University College of Medicine, Busan 49267, Korea; duggymdc@gmail.com; 10Division of Cardiology, Department of Internal Medicine, Wonju Severance Christian Hospital, Yonsei University Wonju College of Medicine, Wonju 26426, Korea; yoonj@yonsei.ac.kr; 11Division of Cardiology, Department of Internal Medicine, Heart Vascular Stroke Institute, Samsung Medical Center, Sungkyunkwan University School of Medicine, Seoul 06351, Korea; cardiokh@gmail.com (K.H.C.); youngbien.song@samsung.com (Y.B.S.); hcgwon62@gmail.com (H.-C.G.); 12Division of Cardiology, Department of Internal Medicine, Yeungnam University Medical Center, Daegu 42415, Korea; pjs@med.yu.ac.kr; 13Division of Cardiology, Department of Internal Medicine, Severance Cardiovascular Hospital, Yonsei University College of Medicine, Seoul 03722, Korea; MKHONG61@yuhs.ac; 14Division of Cardiology, Department of Internal Medicine, Inje University Ilsan Paik Hospital, Goyang 10380, Korea; joon.doh@gmail.com; 15Division of Cardiology, Department of Internal Medicine, Pusan National University Hospital, Busan 49241, Korea; chakws1@hanmail.net; 16Division of Cardiology, Department of Internal Medicine, Inje University Haeundae Paik Hospital, Busan 48108, Korea; jo1216@inje.ac.kr; 17Division of Cardiology, Department of Internal Medicine, Chungbuk National University College of Medicine, Cheongju 28644, Korea; louisahj@gmail.com; 18Division of Cardiology, Department of Internal Medicine, Seoul St. Mary’s Hospital, The Catholic University of Korea, Seoul 06591, Korea; kiyuk@catholic.ac.kr (K.C.); hbhmac@naver.com (B.-H.H.); 19Division of Cardiology, Department of Internal Medicine, Ajou University Hospital, Suwon 16499, Korea; sychoimd@ajou.ac.kr; 20Division of Cardiology, Department of Internal Medicine, Chonnam National University Hospital, Gwangju 61469, Korea; myungho@chollian.net

**Keywords:** coronary bifurcation stenting, predictor, clinical outcome, drug-eluting stents

## Abstract

Background: No large-scale study has compared the clinical and angiographic predictors of cardiovascular events in patients with left main bifurcation (LMB) and non-LMB stenting after second-generation DES implantation. Herein, we investigated differential clinical and angiographic factors for predicting outcomes in LMB versus non-LMB stenting. Methods: A total of 2648 patients with bifurcation lesions treated with second-generation DESs from the retrospective patient cohort were divided into an LMB group (*n* = 935) and a non-LMB group (*n* = 1713). The primary outcome was the 7-year incidence of target lesion failure (TLF), defined as the composite of cardiac death, myocardial infarction, and target lesion revascularization. Results: The incidence of TLF was 9.8%. Those in the LMB group were associated with a higher risk of TLF (14.2% versus 7.5%, *p* < 0.001) than those in the non-LMB group. Regarding the LMB group, independent predictors of TLF were chronic kidney disease (CKD), reduced left ventricular ejection fraction (LVEF), and two-stenting. Regarding the non-LMB group, CKD, reduced LVEF, old age, diabetes, and small diameter of the main vessel stent were independent predictors of TLF. Conclusions: The two-stent strategy could potentially increase TLF for the LMB lesions, and achieving the maximal diameter of the main vessel stent could result in better clinical outcomes for non-LMB lesions.

## 1. Introduction

Coronary bifurcation percutaneous coronary intervention (PCI) has been considered a predictor of restenosis after stent implantation. With the introduction of second-generation drug-eluting stents (DESs), the angiographic rates of restenosis after PCI have been reduced dramatically [1,2,3]. In addition, second-generation DESs have revealed better clinical outcomes in complex stenting techniques for bifurcation lesions than first-generation DESs [4,5]. However, even with second-generation DES, patients with coronary bifurcation PCI have shown increased rates of restenosis and late loss index compared to patients without coronary bifurcation PCI [3,5]. To date, no large-scale study has compared the clinical and angiographic predictors of cardiovascular events in patients with left main bifurcation (LMB) and non-LMB stenting with second-generation DESs. The objectives of this multicenter observational retrospective cohort study were to investigate the differential clinical and angiographic factors involved in predicting the outcomes of left main versus non-left main coronary bifurcation stenting with second-generation DESs.

## 2. Materials and Methods

### 2.1. Study Population

The coronary bifurcation stent III registry is a multicenter observational study with retrospective national cohorts of patients with bifurcation lesions who underwent PCI with second-generation DESs (ClinicalTrials.gov NCT03068494). The retrospective patient cohort consisted of 2648 patients treated between January 2010 and December 2014 in 21 Korean hospitals. The current bifurcation registry design has been described in detail previously [5]. In brief, the coronary bifurcation stent III registry is a real-world registry. The inclusion criteria are as follows: (1) age > 19 years old, (2) any type of coronary bifurcation lesion in the major epicardial artery treated solely with second-generation DESs, and (3) a main vessel (MV) diameter ≥ 2.5 mm and a side branch (SB) diameter ≥ 2.3 mm, confirmed by core laboratory quantitative coronary angiography analysis. Meanwhile, patients who experienced cardiogenic shock or cardiopulmonary resuscitation during hospitalization, protected left main disease, or experienced severe left ventricular systolic dysfunction (ejection fraction < 30%) were excluded from this registry. The study protocol was approved by the institutional review board of each hospital and was conducted according to the principles of the Declaration of Helsinki. The institutional review boards of the participating hospitals waived the requirement for informed consent owing to the retrospective nature of the study.

### 2.2. Percutaneous Coronary Bifurcation Intervention

Index PCI was conducted according to the relevant standard guidelines at the time of each procedure. All the patients received loading doses of aspirin (300 mg) and P2Y_12_ inhibitors (clopidogrel 300–600 mg, prasugrel 60 mg, or ticagrelor 180 mg) before PCI, unless they had previously received the aforementioned antiplatelet medications. Low-molecular-weight or unfractionated heparin was used for anticoagulation to achieve an activated clotting time of 250–300 s during PCI. All the procedures relied on the operator’s discretion regarding the approach strategy, final kissing ballooning (FKB), proximal optimization technique (POT), or re-POT, access site, type of DES, use of glycoprotein IIb/IIIa inhibitors, and use of intravascular imaging or invasive physiological assessments. Although aspirin was prescribed indefinitely after PCI, the maintenance duration of other antiplatelet agents, including clopidogrel, prasugrel, and ticagrelor, was decided at the discretion of the operator.

### 2.3. Data Collection and Quantitative Coronary Angiography Analysis

A web-based reporting system was used to collect patient information, including demographic, medication, laboratory, angiographic, and procedural data. The follow-up clinical outcomes were collected by electronic medical records during the outpatient clinic. Telephone interviews were also used if patients died during the follow-up period. For quantitative coronary angiography analysis, an angiographic core laboratory (Heart Vascular Stroke Institute, Samsung Medical Center, Seoul, Korea) with a validated automated edge-detection system (Centricity CA 1000, GE, Waukesha, WI, USA) reviewed and analyzed all baseline and procedural coronary angiograms. The Medina classification was used for the classification of bifurcation lesions. True bifurcation lesions were defined as Medina classification types 1.1.1, 1.0.1, and 0.1.1. Quantitative coronary angiography analysis was performed for both pre- and post-procedures, bifurcation angle (defined as the angle between the distal MV and the SB at its origin using the angiographic projection with the widest separation of the two branches), minimum lumen diameter, reference vessel diameter, and lesion length for each vessel were measured, and percent diameter stenosis (100 × [reference vessel diameter/minimum lumen diameter]/reference vessel diameter) for each vessel was determined (Appendix A).

### 2.4. Primary and Secondary Outcomes

The primary outcome was the 7-year incidence of target lesion failure (TLF), defined as the composite of cardiac death, target vessel myocardial infarction (MI), and target lesion revascularization (TLR). The secondary outcomes included the individual components of the primary outcome. All the clinical events were verified by an independent clinical event adjudicating committee, composed of independent experts in interventional cardiology who had not participated in patient enrollment. All deaths were considered to be of cardiac cause, unless an undisputed noncardiac cause could be established. Target vessel MI was defined as index procedure-related MI diagnosed by invasive coronary angiography. TLR was defined as a repeat PCI of the lesion within 5 mm of stent deployment.

### 2.5. Statistical Analysis

Continuous variables are presented as mean ± standard deviation and were compared using a Student’s t-test for parametric data and a Mann–Whitney test for nonparametric data. Categorical variables are presented as numbers (percentages) and were compared using a chi-square test or a Fisher’s exact test. The cumulative incidences of clinical events were presented as Kaplan–Meier estimates and were compared using a log-rank test. The patients were censored at 7 years (2555 days) or when events occurred. Hazard ratios (HRs) and 95% CIs were calculated using Cox proportional hazard models, and the proportional hazards assumptions of the HRs in the Cox proportional hazards models were graphically inspected in the log minus log plot and were also tested by Schoenfeld residuals. In multivariable models, the covariates that were either statistically significant in univariate analyses or clinically relevant were considered candidate variables. Adjusted HRs were acquired by Cox regression based on age (≥75 years), sex, hypertension, diabetes mellitus, hyperlipidemia, chronic kidney disease, history of MI, history of PCI, history of stroke, acute coronary syndrome, reduced left ventricular ejection fraction (LVEF) (EF < 40%), LMB, two-stent technique, true bifurcation, transradial intervention, intravascular ultrasound guidance, main vessel maximal stent size, main vessel stent length, FKB, POT, re-POT, noncompliant balloon use, bifurcation angle, use of β-blockers, use of angiotensin-converting enzyme inhibitor or angiotensin receptor blockers, and use of nitrate. Inverse-probability weighting was performed using generalized boosted models for multiple treatments to evaluate the interactions between the LMB and non-LMB groups that could affect the clinical outcomes [6]. Stratified and inverse-probability weighting-adjusted Cox proportional hazard models were used to compare the outcomes of the matched groups. All probability values were two-sided, and *p* values <0.05 were considered statistically significant. Statistical analyses were performed using R Statistical Software (version 3.6.0; R Foundation for Statistical Computing, Vienna, Austria).

## 3. Results

### 3.1. Baseline Clinical Characteristics

The baseline clinical characteristics are presented in Table 1. The mean age was 63.7 ± 11.0 years. Elderly patients, defined as those aged ≥75 years, were 17.4% of the total patients. The majority of the patients were men (76.0%) with comorbidities, such as DM (34.2%), hypertension (56.8%), and dyslipidemia (38.1%). A higher number of patients presented with acute coronary syndrome (*n* = 1619, 61.1%) than with stable angina (*n* = 1029, 38.9%). According to the bifurcation level, 935 (35.3%) patients were treated for LMB and 1713 (64.7%) patients were treated for non-LMB. Compared with patients treated with non-LMB, those treated with LMB were older, more frequently non-smokers (non- or ex-smokers), and more likely to have hypertension, diabetes mellitus, chronic kidney disease, a history of PCI, or to have suffered from a stroke. The LVEF was not significantly different between patients treated with non-LMB and those treated with LMB (58.8 ± 9.6 versus 58.3 ± 10.3%, *p* = 0.208).

### 3.2. Lesion and Procedural Characteristics

Overall, the left anterior descending artery (45.6%) was the most commonly affected bifurcated vessel, followed by the left main artery (35.3%), left circumflex artery (13.2%), and right coronary artery (5.9%). The most common stenting strategy was simple crossover stenting (63.6%). The two-stent technique was performed in 17.1% of the total patients. Everolimus-eluting stent (EES) was the most used second-generation DES in the registry (48.2%), followed by the zotarolimus-eluting Stent (ZES) (27.8%) and the biolimus-eluting stent (BES) (19.4%). Compared with the non-LMB group, the LMB group had higher proportions of multivessel disease and treatment with the two-stent strategy, and a lower proportion of true bifurcation. The implanted stent type differed significantly between the two groups. FKB, POT, and re-POT were more frequently performed in the LMB group. Transradial intervention was used less frequently in the LMB group; however, the intravascular ultrasound was used more frequently in the LMB group than in the non-LMB group (Table 2).

### 3.3. Clinical Outcomes

The median follow-up duration was 53 months (interquartile range, 37–68 months). The total incidence of 7-year TLF was 9.8% in the registry. According to the bifurcation site, the LMB group was associated with a higher risk of TLF (14.2% versus 7.5%, *p* < 0.001) than the non-LMB group. The incidence of target vessel MI (2.3% versus 1.1%, *p* = 0.002), TLR (9.6% versus 3.6%, *p* < 0.001), and all-cause death (8.3% versus 6.9%, *p* = 0.026) was significantly higher in the LMB group than it was in the non-LMB group (Figure 1). However, no difference was found in cardiac death between the two groups (4.7% versus 3.3%, *p* = 0.064) (Figure 1). These results were consistently observed even after inverse-probability weighting adjustment.

### 3.4. Independent Predictors for TLF

In the patients who underwent PCI for bifurcation lesions using second-generation DESs, the Cox regression multivariate analysis revealed that LMB was a significant independent predictor of TLF (adjusted HR 2.06, 95% CI 1.42–2.99, *p* < 0.001). In addition, age (≥75 years) (adjusted HR 1.47, 95% CI 1.02–2.10, *p* = 0.038), chronic kidney disease (adjusted HR 3.06, 95% CI 1.88–4.96, *p* < 0.001), and reduced LVEF (adjusted HR 3.07, 95% CI 1.91–4.93, *p* < 0.001) were independent predictors of TLF. However, the two-stent technique was not a significant predictor of TLF in the overall patient group (adjusted HR 1.54, 95% CI 0.97–2.45, *p* = 0.068) (Table 3).

Regarding the LMB group, the two-stent technique was an independent predictor of TLF (adjusted HR 2.01, 95% CI 1.11–3.64, *p* = 0.021). Chronic kidney disease (adjusted HR 4.06, 95% CI 2.15–7.66, *p* < 0.001) and reduced LVEF (adjusted HR 3.05, 95% CI 1.55–5.99, *p* = 0.001) were also independent predictors of TLF. Regarding the non-LMB group, the two-stent technique was not a significant predictor of TLF. However, age (≥75 years) (adjusted HR 2.21, 95% CI 1.33–3.67, *p* = 0.002), diabetes mellitus (adjusted HR 1.71, 95% CI 1.09–2.69, *p* = 0.020), chronic kidney disease (adjusted HR 2.49, 95% CI 1.13–5.48, *p* = 0.024), reduced LVEF (adjusted HR 3.67, 95% CI 1.83–7.34, *p* < 0.001), and maximal diameter of the main vessel stent (adjusted HR 0.72, 95% CI 0.54–0.96, *p* = 0.024) were predictors of TLF (Table 3).

## 4. Discussion

The main findings of this study were as follows: (1) The coronary bifurcation stent III registry is an all-comer national registry including bifurcation lesions using second-generation DESs. During a 7-year follow-up period, the cumulative incidence of TLF, target vessel MI, and TLR were 9.8%, 1.5%, and 5.7%, respectively. (2) The independent predictors of the 7-year follow-up TLF in the patients using second-generation DES for coronary bifurcation lesions were age (≥75 years old), chronic kidney disease, reduced LVEF (<40%), and LMB. (3) In the LMB group, the two-stent technique, as well as chronic kidney disease and reduced LVEF, were significant predictors of TLF. (4) In the non-LMB group, age ≥ 75 years, diabetes mellitus, chronic kidney disease, reduced LVEF, and the maximal diameter of the main vessel stent were significant predictors of TLF.

This study included 2486 patients who were treated with second-generation DES for bifurcation lesions. Several randomized controlled trials (RCTs) have reported the superiority of second-generation DESs compared with first-generation DESs in terms of clinical and angiographic outcomes [1,2]. In the same context, second-generation DESs have shown better clinical outcomes for bifurcation lesions compared with first-generation DES [7]. The results of the previous coronary bifurcation stent II registry, including both first-generation and second-generation DESs, showed a 10.2% TLF incidence over the entire study period (median 36 months) [8]. In the present coronary bifurcation stent III registry, the overall incidence of TLF was much lower (6.9%), even with a longer follow-up duration (median 53 months). The recently reported RCTs for bifurcation treatment using second-generation DESs focused on a comparison of stent strategies (one stent versus two stents) [9,10,11,12]. Since the current guidelines, which recommend a provisional strategy for the one-stenting technique only in the main vessel as an initial approach for bifurcation lesions, were based on the numerous RCTs using first-generation stents, these previously reported results do not reflect current real-world practices [13,14].

Results of studies comparing one- with two-stenting techniques using second-generation DESs for coronary bifurcation lesions showed vast discrepancies. In the RESOLUTE ALL COMERS trial for 382 patients and the bifurcation substudy of the SPIRIT V for 492 patients, the rate of cardiovascular events was significantly higher in the two-stent technique, even with second-generation DESs [9,15,16]. In contrast, some studies reported favorable outcomes of the two-stenting technique compared to one-stent technique. The Nordic-Baltic Bifurcation Study IV involving 450 patients and the DKCRUSH-II study involving 370 patients reported that the two-stenting technique was comparable to or better than the one-stenting technique [10,17]. In addition, the recently reported two-stenting strategy using the second-generation DES showed no significant differences in the clinical outcomes when compared to the one-stent technique in overall patients with coronary bifurcation [5]. Although there may be several reasons for the discrepancies across various studies, a relatively small number of patients and a substantial proportion of LMB or true bifurcation lesions might be potential confounders that diluted the true differences between one- and two-stenting strategies. In this regard, we investigated long-term clinical outcomes and predictors from our nationwide coronary bifurcation registry, the largest sample size of an unrestricted population that reflects the real-world practice of bifurcation PCI.

In the present coronary bifurcation stent registry, the independent predictors of TLF in second-generation DESs were patient- or lesion-related factors, such as age (≥75 years), chronic kidney disease, reduced LVEF, or a LMB lesion, but not the procedural techniques, including FKI, POT, and two-stenting strategies. These results imply that optimal patient selection and medical treatment for comorbidities are the main factors for successful coronary bifurcation PCI, rather than the procedural-related factors. It should be noted that an LMB lesion is an independent predictor of TLF in patients who underwent PCI using second-generation DESs. Previously, Lee et al. reported that an LMB lesion was a significant predictor of TLF from a registry including first and second-generation DESs [4]. From our most recent coronary bifurcation registry, the risk of target vessel MI or TLR was significantly higher in the LMB group than in the non-LMB group (Figure 1). There are several plausible explanations for the adverse outcomes of LMB stenting. First, the left main artery plays a crucial role in supplying most of the left ventricular region [18]. Thus, it should be noted that occlusion of the SB (left circumflex artery) translates into severe adverse clinical outcomes. Another explanation is the geometric uniqueness of the LMB, including its short length, the common prevalence of trifurcated vessels, large-sized parent vessels with a rapidly tapering MV caliber, and wider bifurcation angle [19]. Our results showed that the two-stenting strategy was conducted more frequently in the LMB group than in the non-LMB group (27.1% versus 11.7%, *p* < 0.001), demonstrating the clear intention of maintaining good patency at left circumflex artery.

In the LMB group, as observed in all the subjects, chronic kidney disease and reduced LVEF were significant predictors of TLF. In addition, although IVUS-guided PCI was more frequently conducted in the LMB group, the use of IVUS was not a significant predictor of TLF in our study. The plausible explanation is that the LMB lesion as a strong independent predictor might partially negate the potential benefit of IVUS use for bifurcation lesions. Of note, the two-stenting strategy for LMB lesions led to more frequent adverse clinical outcomes compared with the one-stenting strategy. The optimal PCI strategy for LM bifurcation lesions is still debatable. Similar to our study, a substudy of the EXCEL trial reported that the two-stenting strategy in the LMB was associated with more adverse clinical outcomes than the one-stent technique with a provisional strategy [20]. In contrast, the DKCRUSH-V study reported less incidence of 1-year TLF in the two-stenting group when compared to the one-stent group with a provisional strategy in the LMB [21]. In randomized controlled trials, such as the DKCRUSH-V study, patients undergoing LMB PCI are not representative of the real-world population. Our results from the coronary bifurcation national registry, which represents a large-sized all-comer bifurcation population with long-term follow-up, provide robust clinical implications for real-world coronary bifurcation PCI.

In the non-LMB group, diabetes mellitus and not achieving maximal diameter of the main vessel stent were significant predictors of TLF. Diabetes mellitus plays a crucial role in various pathophysiological mechanisms in coronary artery disease with respect to vascular inflammation, endothelial dysfunction, and dysregulation of growth factor expression [22]. Thus, diabetes mellitus is a significant predictor of coronary restenosis, even in the DES era [23], especially for small main vessel stent diameter in the non-LMB group in our study. Thus, obtaining maximal diameter of the main vessel stent would be more crucial for non-LMB lesions in comparison to LMB lesions.

### Limitations

There are a few limitations to this study. First, due to the nature of the retrospective study, our results may have been affected by unidentified confounding factors; however, we have tried to reduce the potential confounders by multiple sensitivity analysis, including multivariable Cox regression and inverse-probability weighting. Second, duration and type of dual antiplatelet therapy were not available in our data. Finally, patients with cardiogenic shock or cardiopulmonary resuscitation during hospitalization, protected left main disease, and severe left ventricular systolic dysfunction were excluded from our registry.

## 5. Conclusions

Treatment with a second generation stent in the bifurcation lesion reduced the overall incidence of TLR. Regarding LMB, a two-stent strategy could potentially increase TLF, irrespective of stent diameter. Meanwhile, in non-LMB, old age, diabetes, and small main vessel stent diameter could potentially increase TLF.

## Figures and Tables

**Figure 1 jcm-10-03024-f001:**
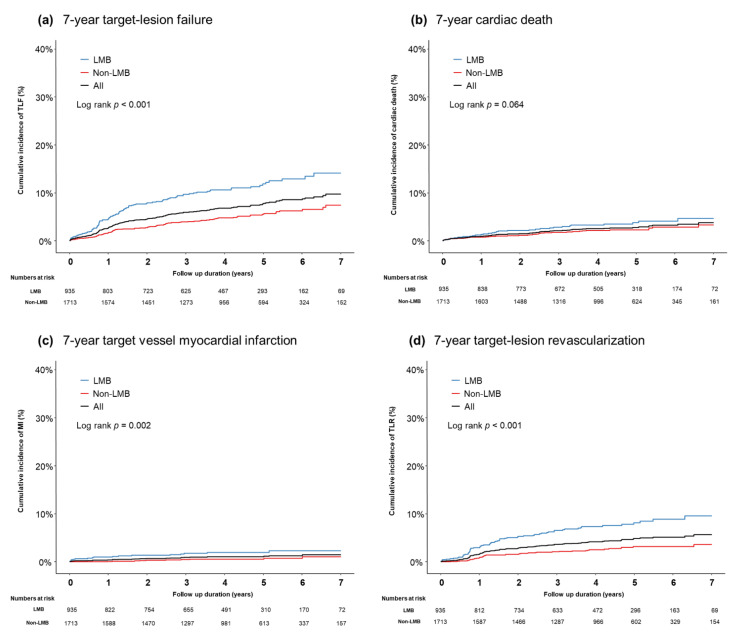
A cumulative incidence of clinical outcomes according to the bifurcation location: (**a**) 7-year TLF, (**b**) 7-year cardiac death, (**c**) 7-year target vessel MI, and (**d**) 7-year TLR.

**Table 1 jcm-10-03024-t001:** Baseline clinical characteristics.

	Overall	Non-LMB	LMB	*p*
	(*n* = 2648)	(*n* = 1713)	(*n* = 935)	
Age (years)	63.7 ± 11.0	62.8 ± 11.1	65.5 ± 10.7	<0.001
The elderly (≥75 years old), *n* (%)	462 (17.4)	278 (16.2)	184 (19.7)	0.029
Male, *n* (%)	2013 (76.0)	1302 (76.0)	711 (76.0)	>0.999
Hypertension, *n* (%)	1504 (56.8)	947 (55.3)	557 (59.6)	0.037
Diabetes mellitus, *n* (%)	905 (34.2)	549 (32.0)	356 (38.1)	0.002
Dyslipidemia, *n* (%)	1009 (38.1)	648 (37.8)	361 (38.6)	0.724
Chronic Kidney Disease, *n* (%)	103 (3.9)	56 (3.3)	47 (5.0)	0.033
Current smoker, *n* (%)	798 (30.1)	572 (33.4)	226 (24.2)	<0.001
Family Hx of CAD, *n* (%)	83 (3.1)	56 (3.3)	27 (2.9)	0.673
Previous MI, *n* (%)	113 (4.3)	65 (3.8)	48 (5.1)	0.126
Previous PCI, *n* (%)	323 (12.2)	164 (9.6)	159 (17.0)	<0.001
Previous stroke, *n* (%)	177 (6.7)	101 (5.9)	76 (8.1)	0.034
Clinical presentations				0.272
Stable angina, *n* (%)	1029 (38.9)	652 (38.1)	377 (40.3)	
Acute coronary syndrome, *n* (%)	1619 (61.1)	1061 (61.9)	558 (59.7)	
LV ejection fraction (%)	58.6 ± 9.9	58.8 ± 9.6	58.3 ± 10.3	0.208
Reduced LV function (LVEF ≤ 40%), *n* (%)	117 (4.4)	66 (3.9)	51 (5.5)	0.069

LMB, left main bifurcation; CAD, coronary artery disease; MI, myocardial infarction; PCI, Percutaneous coronary intervention; LVEF, left ventricular ejection fraction.

**Table 2 jcm-10-03024-t002:** Baseline lesion and procedural characteristics.

	Overall	Non-LMB	LMB	*p*
	(*n* = 2648)	(*n* = 1713)	(*n* = 935)	
Multi-vessel disease, *n* (%)	1647 (62.2)	873 (51.0)	774 (82.8)	<0.001
Location of bifurcated vessel, *n* (%)				<0.001
LAD	1208 (45.6)	1208 (70.5)	0 (0.0)	
LCX	350 (13.2)	350 (20.4)	0 (0.0)	
LM	935 (35.3)	0 (0.0)	935 (100.0)	
RCA	155 (5.9)	155 (9.0)	0 (0.0)	
Medina classification, *n* (%)				<0.001
0.0.1	95 (3.6)	39 (2.3)	56 (6.0)	
0.1.0	575 (21.7)	291 (17.0)	284 (30.4)	
0.1.1	248 (9.4)	175 (10.2)	73 (7.8)	
1.0.0	296 (11.2)	212 (12.4)	84 (9.0)	
1.0.1	168 (6.3)	130 (7.6)	38 (4.1)	
1.1.0	427 (16.1)	248 (14.5)	179 (19.1)	
1.1.1	839 (31.7)	618 (36.1)	221 (23.6)	
True bifurcation, *n* (%)	1255 (47.4)	923 (53.9)	332 (35.5)	<0.001
Stent Technique, *n* (%)				<0.001
Simple Crossover	1685 (63.6)	1196 (69.8)	489 (52.3)	
1 stent with SB Balloon	475 (17.9)	300 (17.5)	175 (18.7)	
Classic Crush	99 (3.7)	32 (1.9)	67 (7.2)	
Balloon Crush	65 (2.5)	37 (2.2)	28 (3.0)	
Mini Crush	80 (3.0)	33 (1.9)	47 (5.0)	
Culotte	31 (1.2)	15 (0.9)	16 (1.7)	
TAP	74 (2.8)	30 (1.8)	44 (4.7)	
Classic T	51 (1.9)	35 (2.0)	16 (1.7)	
Kissing Stent	41 (1.5)	15 (0.9)	26 (2.8)	
Others	47 (1.8)	20 (1.2)	27 (2.9)	
2-stenting strategy, *n* (%)	454 (17.1)	201 (11.7)	253 (27.1)	<0.001
No. of stents	1.79 ± 0.97	1.72 ± 0.95	1.93 ± 1.00	<0.001
Transradial approach, *n* (%)	1507 (56.9)	1035 (60.4)	472 (50.5)	<0.001
IVUS-guidance PCI, *n* (%)	1054 (39.8)	460 (26.9)	594 (63.5)	<0.001
Rotablator, *n* (%)	16 (0.6)	5 (0.3)	11 (1.2)	0.011
Cutting balloon, *n* (%)	50 (1.9)	6 (0.4)	44 (4.7)	<0.001
NC balloon, *n* (%)	534 (20.2)	285 (16.6)	249 (26.6)	<0.001
FKB, *n* (%)	789 (29.8)	393 (22.9)	396 (42.4)	<0.001
POT, *n* (%)	739 (27.9)	446 (26.0)	293 (31.3)	0.004
Re-POT, *n* (%)	123 (4.6)	50 (2.9)	73 (7.8)	<0.001
Maximal diameter of MV stents (mm)	3.13 ± 0.65	2.99 ± 0.62	3.39 ± 0.62	<0.001
Minimal diameter of MV stents (mm)	3.05 ± 0.63	2.93 ± 0.61	3.26 ± 0.62	<0.001
Cumulative length of MV stents (mm)	28.89 ± 13.66	28.55 ± 12.63	29.51 ± 15.35	0.101

LMB, left main bifurcation; LAD, left anterior descending; LCX, left circumflex; LM, left main; RCA, right coronary artery; IVUS, intravascular ultrasound; PCI, percutaneous coronary intervention; NC, non-compliant; FKB, final kissing balloon; POT, proximal optimization technique; MV, main vessel; SB, side branch.

**Table 3 jcm-10-03024-t003:** Independent predictor of 7-year target lesion failure.

	Crude HR	Adjusted HR	95% CI	*p* Value
All patients (*n* = 2648)				
The elderly (≥75 years old)	1.60	1.47	1.02–2.10	0.038
Chronic kidney disease	4.35	3.06	1.88–4.96	<0.001
Reduced LVEF (EF ≤ 40%)	3.51	3.07	1.91–4.93	<0.001
Left main bifurcation PCI	2.24	2.06	1.42–2.99	<0.001
Two-stenting strategy (vs. provisional)	1.92	1.54	0.97–2.45	0.068
Left main bifurcation group (*n* = 935)				
Chronic kidney disease	4.41	4.06	2.15–7.66	<0.001
Reduced LVEF (EF ≤ 40%)	2.98	3.05	1.55–5.99	<0.001
Two-stenting strategy (vs provisional)	1.88	2.01	1.11–3.64	0.021
Non-left main bifurcation group (*n* = 1713)				
The elderly (≥75 years old)	2.30	2.21	1.33–3.67	0.002
Diabetes mellitus	1.97	1.71	1.09–2.69	0.020
Chronic kidney disease	3.71	2.49	1.13–5.48	0.024
Reduced LVEF (EF ≤ 40%)	3.87	3.67	1.83–7.34	<0.001
Two-stenting strategy (vs provisional)	1.17	1.15	0.50–2.63	0.738
Maximal diameter of MV stents (mm)	0.73	0.72	0.54–0.96	0.024

HR, hazard ratio; LVEF, left ventricular ejection fraction; MV, main vessel; PCI, percutaneous coronary intervention.

## Data Availability

The data generated in this study are available from the corresponding author(s) upon reasonable request.

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
