# Peer review of "Differential Factors for Predicting Outcomes in Left Main versus Non-Left Main Coronary Bifurcation Stenting"

_jcm, 2021, doi:10.3390/jcm10143024_

Round 1

Reviewer 1 Report

Cha et al present a large retrospective study on LM vs non-LM bifurcation stenting outcomes and the factors affecting these outcomes. I have a couple comments that require adressing:

-definition of TLF is quite broad: MI (definition ck-mbm OR troponin elevation with symptoms OR ecg-findings suggestive of ischemia), TLR or cardiac death. As a longer follow-up is considered to be an advantage in many aspects, there are concerns that over the years the MIs are not related to the index bifurcation stenosis. Considering the somewhat vague definition of MI, and as the main result of TLF is driven mainly by TLR and MI (and not cardiac death), the connection between index procedure and outcome event is not intuitively clear. Also, LMB patients had significantly more often a multivessel disease status (82.8% vs 51%), which further arises the question of prossible de novo lesions causing MIs? Did these Mi patients undergo ICA to identify the culprit lesion?

-When identifying the predictors of TLF, which variables were included in the multivariable analysis?

-As the authors mention adjusted HRs - which factors were adjusted? All significantly differing baseline variables?

Author Response

In Reply to Reviewer 1

We deeply appreciate your time and input. We have carefully considered your comments and have addressed them. The revised parts in the manuscript are expressed in yellow highlight.

[Response to Remarks]

  1. Reviewer commented:

Cha et al present a large retrospective study on LM vs non-LM bifurcation stenting outcomes and the factors affecting these outcomes. I have a couple comments that require addressing:

-definition of TLF is quite broad: MI (definition ck-mbm OR troponin elevation with symptoms OR ecg-findings suggestive of ischemia), TLR or cardiac death. As a longer follow-up is considered to be an advantage in many aspects, there are concerns that over the years the MIs are not related to the index bifurcation stenosis. Considering the somewhat vague definition of MI, and as the main result of TLF is driven mainly by TLR and MI (and not cardiac death), the connection between index procedure and outcome event is not intuitively clear. Also, LMB patients had significantly more often a multivessel disease status (82.8% vs 51%), which further arises the question of possible de novo lesions causing MIs? Did these Mi patients undergo ICA to identify the culprit lesion?

Authors’ response:

We thank the reviewer for the insightful comments. And we apologize for the confusion. As the reviewer commented, the connection between index procedure and outcome event is important to interpret this study. In this study, myocardial infarction (MI) events were investigated in terms of spontaneous MI (an elevation of the creatine kinase-myocardial band or a troponin level greater than the upper normal limit with concomitant ischemic symptoms or electrocardiography findings indicative of ischemia that was not related to the index procedure) and target vessel MI (index procedure-related MI). Since the connection between index procedure and outcome event is important, target lesion failure, as the primary endpoint, was a composite of cardiac death, target vessel MI, and target lesion revascularization, not included spontaneous MI. All target vessel MI was confirmed by invasive coronary angiography to find a correlation with index procedure. Thus, we have revised the manuscript and figure legend in response to the reviewer’s comment.

Materials and Methods, page 3

2.4. Primary and secondary outcomes

The primary outcome was the 7-year incidence of target lesion failure (TLF), defined as the composite of cardiac death, target vessel myocardial infarction (MI), and target lesion revascularization (TLR). The secondary outcomes included the individual components of the primary outcome. All the clinical events were verified by an independent clinical event adjudicating committee, composed of independent experts in interventional cardiology who had not participated in patient enrollment. All deaths were considered to be of cardiac cause unless an undisputed noncardiac cause could be established. Target vessel MI was defined as index procedure-related MI diagnosed by invasive coronary angiography. TLR was defined as a repeat PCI of the lesion within 5 mm of stent deployment.

Results, page 7

3.3. Clinical outcomes

The incidence of target vessel MI (2.3% versus 1.1%, p=0.002), TLR (9.6% versus 3.6%, p<0.001), and all-cause death (8.3% versus 6.9%, p=0.026) were significantly higher in the LMB group than in the non-LMB group (Fig. 1.)

Discussion, page 9

During a 7-year follow-up period, the cumulative incidence of TLF, target vessel MI, and TLR were 9.8%, 1.5%, and 5.7%, respectively.

Figure legend, page 8

Figure 1. A Cumulative Incidence of Clinical Outcomes According to the Bifurcation Location. (a) 7-year TLF, (b) 7-year cardiac death (c) 7-year target vessel MI, and (d) 7-year TLR.

  1. Reviewer commented:

-When identifying the predictors of TLF, which variables were included in the multivariable analysis?

Authors’ response:

We thank the reviewer for the comments. In this study, we performed multivariable analysis using covariates that were either clinically relevant or statistically significant in univariate analyses. As described in Materials and Methods, such variables included age (≥75 years), sex, hypertension, diabetes mellitus, hyperlipidemia, chronic kidney disease, history of MI, history of PCI, history of stroke, acute coronary syndrome, reduced LVEF (EF <40%), LMB, two-stent technique, true bifurcation, transradial intervention, intravascular ultrasound guidance, main vessel maximal stent size, main vessel stent length, FKB, POT, re-POT, noncompliant balloon use, bifurcation angle, use of β-blockers, use of angiotensin-converting enzyme inhibitor or angiotensin receptor blockers, and use of nitrate.

  1. Reviewer commented:

-As the authors mention adjusted HRs - which factors were adjusted? All significantly differing baseline variables?

Authors’ response:

We thank the reviewer for the comments. As the reviewer commented, we performed multivariable analysis for acquired adjusted HRs using covariates that were either clinically relevant or statistically significant in univariate analyses. Same as the response to the reviewer’s second comment, such variables included age (≥75 years), sex, hypertension, diabetes mellitus, hyperlipidemia, chronic kidney disease, history of MI, history of PCI, history of stroke, acute coronary syndrome, reduced LVEF (EF <40%), LMB, two-stent technique, true bifurcation, transradial intervention, intravascular ultrasound guidance, main vessel maximal stent size, main vessel stent length, FKB, POT, re-POT, noncompliant balloon use, bifurcation angle, use of β-blockers, use of angiotensin-converting enzyme inhibitor or angiotensin receptor blockers, and use of nitrate. In subgroup analysis for LMB and non-LMB, one covariate (LMB) was excluded from the multivariable analysis.

Reviewer 2 Report

Dr Cha and colleagues performed a retrospective multicenter observational study of the coronary bifurcation stent III registry and analysed differential factors for predicting outcomes in left main versus non-left main coronary bifurcation stenting.

This is a unique large cohort of patients treated for bifurcation disease. The authors are to be congratulated for setting up this registry and for the detailed follow-up of the patients. The research question is certainly of interest and clinical relevance.

Several grammatical errors

  • Line 82 : replace “Till date” with “To date”
  • Page 9 last paragraph: delete “authors”.

Several observations

  • The 7 year TLR rates are very follow, even keeping in mind that the number of patients with 7 year FU are small. Could this be because a large proportion of patients in the NON-LM group had smaller (unimportant) side branches? I see there was no QCA performed/provided. I wonder what the mean diameter of the SB vessel was?
  • Could you supply data on the stent sizes in the SB in the 2-stent groups?
  • Any data on incidence of stent thrombosis?
  • POT was only performed in 27.9 % of all bifurcations and KBI only in 29.8 % of cases. These are very low. Even in LM bifurcations POT was performed in only 31.3 % of cases. Why is this? All bifurcation PCI guidelines recommend strongly to perform POT, and frequently also FKB? One reason might be that the SB were quite small and thus POT wasn’t really necessary? Please explain.
  • Why do you think there was no difference in the IVUS guided group? Up to 60% IVUS use in the LM. Is this because of the non-randomized fashion? In most large studies of complex PCI (including bifurcations) IVUS guided PCI reduces MACE rates.
  • Please clarify better what is meant with “obtaining the maximal diameter of the main vessel stent was a protective predictor of TLF in the non-LMB goup” (page 9). Or the following statement on page 10 “…..not achieving maximal diameter of the main vessel stent were significant predictors of TLF”. Does this mean that underexpansion was a predictive factor or that the larger the MV stent size the less chance of TLF? If underexpansion of the MV stent, how was this measured?

In methodology QCA was performed. Is the QCA data available? Perhaps a table with QCA data can be provided?

  • The conclusion sentence (page 11) is a bit unclear “Two stent strategy could potentially increase the TLF during 7 year follow-up….. I’m not sure this is the main conclusion. I would certainly also focus on the overall low TLR rates in this cohort of patients

Author Response

In Reply to Reviewer 2

We deeply appreciate your time and input. We have carefully considered your comments and have addressed them.

  1. Reviewer commented:
    Dr Cha and colleagues performed a retrospective multicenter observational study of the coronary bifurcation stent III registry and analysed differential factors for predicting outcomes in left main versus non-left main coronary bifurcation stenting.

This is a unique large cohort of patients treated for bifurcation disease. The authors are to be congratulated for setting up this registry and for the detailed follow-up of the patients. The research question is certainly of interest and clinical relevance.

Several grammatical errors

  • Line 82 : replace “Till date” with “To date”
  • Page 9 last paragraph: delete “authors”.

Authors’ response:

We thank the reviewer for the comments. And we agreed with the reviewer’s comment. So, we have revised the manuscript in response to the reviewer’s comment.

Introduction, page 2

To date, no large-scale study has compared the clinical and angiographic predictors of cardiovascular events in patients with left main bifurcation (LMB) and non-LMB stenting with second-generation DESs.

Discussion, page 10

  1. Discussion

Authors The main findings of this study were as follows:

  1. Reviewer commented:

The 7 year TLR rates are very follow, even keeping in mind that the number of patients with 7 year FU are small. Could this be because a large proportion of patients in the NON-LM group had smaller (unimportant) side branches? I see there was no QCA performed/provided. I wonder what the mean diameter of the SB vessel was?

Authors’ response:

We thank the reviewer for the comments. And we agree with the reviewer’s comment. However, compared to the study, which included both first-generation and second-generation DESs, our results, which used only second-generation DES, showed a reduction of TLF rate even in a longer follow-up duration. As the reviewer commented, due to the nature of coronary anatomy, the side branch diameter of non-LMB was smaller than those of LMB (2.44 ± 0.31 Vs. 2.90 ± 0.48, p<0.001) in QCA analysis. However, our study investigated clinical outcomes after PCI at a side branch (SB) with diameter ≥2.3 mm, which would be eligible for stent implantation.

  1. Reviewer commented:

Could you supply data on the stent sizes in the SB in the 2-stent groups?

Authors’ response:

We thank the reviewer for the comments. In the two-stent strategy group, the stent sizes in the side branch was 2.76 ± 0.15 mm (Non-LMB group vs. LMB group; 2.74 ± 0.11 vs. 2.80 ± 0.21, p<0.001). We have added supplemental table 1 which showed QCA analysis in the manuscript in response to the reviewer’s comment.

Supplemental Table 1. Quantitative coronary angiography analysis

  1. Reviewer commented:

Any data on incidence of stent thrombosis?

Authors’ response:

We thank the reviewer for the comments. Total incidence of stent thrombosis during 7-year was 1.7%. And the LMB group had numerically higher incidence of stent thrombosis than the non-LMB group without significance (3.0% vs 1.1% log rank p=0.085).

  1. Reviewer commented:

POT was only performed in 27.9 % of all bifurcations and KBI only in 29.8 % of cases. These are very low. Even in LM bifurcations POT was performed in only 31.3 % of cases. Why is this? All bifurcation PCI guidelines recommend strongly to perform POT, and frequently also FKB? One reason might be that the SB were quite small and thus POT wasn’t really necessary? Please explain.

Authors’ response:

We thank the reviewer for the comments. And we agreed with the reviewer’s comment. However, this study was conducted between 2010 and 2014. In 2014, the European bifurcation club reported the first consensus for the treatment of bifurcation lesions [1]. During our study periods, only FKB in the two-stent strategy was recommended. In simple stenting, FKB was performed when an angiographically significant (>75% DS or TIMI flow <3) ostial SB lesion remained after MV stenting. Also, the POT or re-POT was performed in cases with large differences in reference diameter between proximal and distal MV. Thus, a future study, investigating the impact of optimization in bifurcation PCI based on the recent PCI guidelines, should be addressed.

  1. Jens Flensted, L.; Niels Ramsing, H.; Goran, S.; Thierry, L.; Alaide, C.; David, H.-S.; Manuel, P.; Olivier, D.; Remo, A.; Miroslaw, F.; et al. Percutaneous coronary intervention for coronary bifurcation disease: consensus from the first 10 years of the European Bifurcation Club meetings. EuroIntervention 2014, 10, 545-560.

  1. Reviewer commented:

Why do you think there was no difference in the IVUS guided group? Up to 60% IVUS use in the LM. Is this because of the non-randomized fashion? In most large studies of complex PCI (including bifurcations) IVUS guided PCI reduces MACE rates.

Authors’ response:

We thank the reviewer for the insightful comments. And we totally agree with the reviewer’s comment. As the reviewer commented, this study was based on the observation prospective registry, not randomized fashion. Thus, the LMB group had a tendency of frequent IVUS use which could be explained by the clinical importance of the left main disease. As a result, the LMB group had higher frequency of stent optimization techniques such as POT, FKB, and/or re-POT (overall 81.5%). Also, various studies reported that IVUS guided PCI reduced the rate of MACE in complex PCI, as the reviewer commented. However, in bifurcation lesions, the LMB lesion as a strong independent predictor might partially negate the potential benefit of IVUS use.

Discussion, page 11

The plausible explanation is that the LMB lesion as a strong independent predictor might partially negate the potential benefit of IVUS use for bifurcation lesions.

  1. Reviewer commented:

Please clarify better what is meant with “obtaining the maximal diameter of the main vessel stent was a protective predictor of TLF in the non-LMB goup” (page 9). Or the following statement on page 10 “…..not achieving maximal diameter of the main vessel stent were significant predictors of TLF”. Does this mean that underexpansion was a predictive factor or that the larger the MV stent size the less chance of TLF? If underexpansion of the MV stent, how was this measured?

Authors’ response:

We apologize for the confusion. And we agree with the reviewer’s comment. Thus, we have revised the manuscript in response to the reviewer’s comment.

Results, page 9

Regarding the non-LMB group, the two-stent technique was not a significant predictor of TLF. However, age (≥75 years) (adjusted HR 2.21, 95% CI 1.33–3.67, p=0.002), diabetes mellitus (adjusted HR 1.71, 95% CI 1.09–2.69, p=0.020), chronic kidney disease (adjusted HR 2.49, 95% CI 1.13–5.48, p=0.024), reduced LVEF (adjusted HR 3.67, 95% CI 1.83–7.34, p<0.001), and maximal diameter of the main vessel stent (adjusted HR 0.72, 95% CI 0.54–0.96, p=0.024) were predictors of TLF (Table 3).

Discussion, page 10

In the non-LMB group, age ≥75 years, diabetes mellitus, chronic kidney disease, reduced LVEF, and the maximal diameter of the main vessel stent were significant predictors of TLF.

  1. Reviewer commented:

In methodology QCA was performed. Is the QCA data available? Perhaps a table with QCA data can be provided?

Authors’ response:

We thank the reviewer for the insightful comments. We have added supplemental table 1 which showed QCA analysis in the manuscript in response to the reviewer’s comment.

Supplemental Table 1. Quantitative coronary angiography analysis

  1. Reviewer commented:

The conclusion sentence (page 11) is a bit unclear “Two stent strategy could potentially increase the TLF during 7 year follow-up….. I’m not sure this is the main conclusion. I would certainly also focus on the overall low TLR rates in this cohort of patients

Authors’ response:

We thank the reviewer for the comments. And we agree with the reviewer’s comment. Thus, we have revised the manuscripts in response to the reviewer’s comment.

Conclusions, page 11

Treated with 2nd generation stent in the bifurcation lesion reduced the overall incidence of TLR. Regarding LMB, a two-stent strategy could potentially increase the TLF irrespective of stent diameter. Meanwhile, in non-LMB, old age, diabetes, and small main vessel stent diameter could potentially increase the TLF.

Round 2

Reviewer 1 Report

I'm satisfied with the answers provided, thank you. Good work.

Reviewer 2 Report

The authors have answered the querries and questions posed.